# Leukemia Incidence by Occupation and Industry: A Cohort Study of 2.3 Million Workers from Ontario, Canada

**DOI:** 10.3390/ijerph21080981

**Published:** 2024-07-27

**Authors:** Konrad Samsel, Tanya Navaneelan, Nathan DeBono, Louis Everest, Paul A. Demers, Jeavana Sritharan

**Affiliations:** 1Occupational Cancer Research Centre, Ontario Health, Toronto, ON M5G 2L3, Canada; 2Dalla Lana School of Public Health, University of Toronto, Toronto, ON M5T 3M7, Canada

**Keywords:** cohort, occupational cancer, surveillance, leukemia subtype, chronic lymphocytic leukemia, acute myeloid leukemia

## Abstract

Although a significant body of evidence has attributed certain occupational exposures with leukemia, such as benzene, formaldehyde, 1,3-butadiene and ionizing radiation, more research is needed to identify work environments at increased risk for this disease. Our study aimed to identify occupational and industry groups associated with an elevated incidence of leukemia using a diverse cohort of workers’ compensation claimants from Ontario, Canada. A total of 2,363,818 workers in the Occupational Disease Surveillance System (ODSS) cohort, with claims between 1983–2019, were followed for malignant leukemia diagnoses up to 31 December 2019. We used a Cox proportional-hazards model to estimate the relative incidence of leukemia in specific occupation and industry groups. After adjusting for age and birth year, males in protective services (HR = 1.17, 95% CI = 1.02–1.35), metal machining (HR = 1.23, 95% CI = 1.07–1.41), transport (HR = 1.15, 95% CI = 1.06–1.25), and mining occupations (HR = 1.28, 95% CI = 1.02–1.60) had elevated risks of leukemia compared to other workers in the ODSS, with comparable findings by industry. Among female workers, slight risk elevations were observed among product fabricating, assembling, and repairing occupations, with other increased risks seen in furniture and fixture manufacturing, storage, and retail industries. These findings underscore the need for exposure-based studies to better understand occupational hazards in these settings.

## 1. Introduction

Despite considerable advancement in treatment, leukemia continues to be characterized by a high mortality rate [1]. While this disease accounts for approximately 2.5% of global cancer diagnoses, similar figures suggest that it is responsible for 3.1% of cancer-attributable deaths worldwide [2]. Among well-studied leukemia subtypes, it is estimated that only 5–10% have a hereditary link [3], underscoring the need for research into behavioral and environmental risk factors to better inform prevention strategies.

Considering the role of environmental factors, the International Agency for Research on Cancer (IARC) and other literature summarizing evidence from observational studies have identified ionizing radiation, 1,3-butadiene (industrial chemical), benzene, and formaldehyde as occupational hazards associated with leukemia [4,5,6]. Occupations at risk of exposure to ionizing radiation include medical imaging, nuclear power plant operations, research laboratories, air travel, and work in the mining, oil, and gas sectors [7]. 1,3-Butadiene and benzene are commonly used in manufacturing and petroleum industries, with airborne exposure also arising from combustion engine emissions [8,9,10,11]. While similar risks of formaldehyde exposure have been noted to arise from manufacturing and petroleum settings, its broader use as an industrial disinfectant also leads to heightened risks of exposure among workers in construction and biomedical research [12]. Among hazards with limited evidence of an association, styrene (industrial chemical) and diazinon (pesticide) are noted for their possible association with this cancer [5]. Taking into account exposure circumstances, IARC has identified synthetic rubber manufacturing as an industry with sufficient evidence for an increased risk of leukemia [5].

Leukemia can be further differentiated into subtypes based on cellular origin and pathology [13]. Its four predominant classifications include acute myeloid leukemia (AML), chronic myeloid leukemia (CML), acute lymphoblastic leukemia (ALL), and chronic lymphocytic leukemia (CLL) [14]. A key distinction between acute and chronic leukemias is that the latter has a higher tendency to be asymptomatic and exhibit a slower disease progression [15,16]. Notably, these four subtypes also differ in their epidemiology and prognosis. Among Canadian adults, CLL diagnoses are most common, accounting for approximately 44% of new leukemia cases [17]. Conversely, AML exhibits a higher mortality relative to other subtypes [18], often making it the subject of large occupational studies [19,20,21]. For certain occupational hazards, the degree of evidence supporting an association varies by leukemia subtype. There is sufficient evidence from epidemiological studies that exposure to benzene can increase the risk of AML [5]. More broadly, there is sufficient evidence linking myeloid leukemias (AML and CML) to formaldehyde [5]. Lastly, there remains limited evidence for the effects of ethylene oxide, an industrial disinfectant, on the incidence of CLL [5].

Prior research into the relationship between occupation and leukemia incidence using administrative data have produced varied findings [19,20,22]. The Nordic Occupational Cancer (NOCCA) Project, which followed 15 million workers for incident cancer diagnosis, found public safety workers, sales agents, and clerical workers to have a higher overall incidence of leukemia compared to others in the cohort [19]. In a similar cohort study of 11 million Korean workers, an increased incidence of leukemia was found among industry groups related to machinery manufacturing, as well as waste collection, treatment, and disposal [22]. Among these prior studies, stratifying results by sex has often yielded differing effect estimates and statistical precision between male and female workers [19,22]. While understanding the role of sex as an effect measure modifier requires further investigation, it has been suggested that physiological differences and varying patterns of exposure to occupational hazards, including differences in frequency and level, may be contributing factors [23]. Given the frequent underrepresentation of female workers in occupational cancer research [23], sex-stratified analyses can help identify risks unique to females in the workforce. This study, therefore, aimed to identify occupation and industry groups associated with an excess risk of leukemia, with a particular focus on differences by sex.

## 2. Materials and Methods

### 2.1. Study Population

The Occupational Disease Surveillance System (ODSS) contains demographic, occupational, and cancer diagnosis information among workers in Ontario, Canada. As previously reported [24], the ODSS was established through the deterministic and probabilistic linkage of workers’ lost time compensation claims data with the Registered Persons Database (RPDB) and the Ontario Cancer Registry (OCR). Accepted lost time claims from 1983 to 2019 (n = 2,387,756) were obtained from Ontario’s Workplace Safety Insurance Board (WSIB), with records including the sex, birth date, and employment of each claimant. Information on employment consisted of each worker’s occupation and industry classification at the time of their lost time claim. Overall, approximately three quarters of Ontario’s workforce receives WSIB coverage [25], with domestic employees, entertainment industry workers, and independent operators exempt from mandatory coverage requirements [26,27]. Cohort entry was determined based on the date of the first-accepted claim, excluding first claims related to cancer diagnoses (prevalent cases). Individuals were also excluded if they were under the age of 15 on their first claim date or if their claim had missing information on sex, birthdate, or both occupation and industry (n = 17,557). The RPDB contains information on death, emigration out of Ontario, and provincial health insurance coverage. Incomplete linkage to the RBDB and OCR resulted in the exclusion of an additional 5710 records. Following the removal of prevalent cases occurring prior to cohort entry (n = 671), the final eligible cohort consisted of 2,363,818 workers. 

### 2.2. Industry, Occupation, and Outcome Classification

Each worker’s occupation and industry were reported at the time of a lost time compensation claim. Occupation was coded by WSIB based on the 1971 Canadian Classification Dictionary of Occupation (CCDO) at the 4-digit level. The first two digits correspond to division-level groups (broadest level of classification), while the third and fourth digits correspond to major- and minor-level groups (detailed classification of the occupation). Industry was coded by WSIB according to the 1970 and 1980 versions of the Canadian Standard Industry Classification (SIC), similarly comprising division, major, and minor levels. While industry results are reported, a greater emphasis was placed on occupation groups, as it is better regarded as a means to represent occupational hazards and other workplace characteristics [28]. Workers with multiple claims during follow-up could be classified into more than one industry or occupation group.

Cancer incidence data, coded according to standards defined by the National Cancer Institute’s Surveillance, Epidemiology, and End Results (SEER) Program, were obtained from the OCR (1964–2019). The OCR’s case definitions were derived from a combination of hospital admission and discharge records, treatment records from patients referred to cancer treatment centers, as well as pathology reports from public hospitals and community laboratories. Incident cases of leukemia were defined by the earliest date of diagnosis with a behavior code of 3 (malignant) and any of the following SEER Site Recode ICD-O-3 definitions [29]: 35021, 35031, 35022, 35011, 35012, 35013, 35023, 35041, and 35043. By extension, subtypes were classified based on the earliest date of diagnosis among the following groups of SEER Site Recode definitions: AML (35021, 35031), CML (35022), ALL (35011), and CLL (35012).

### 2.3. Statistical Analysis

Statistical analysis was conducted using SAS 9.4 (SAS Institute, Cary, NC, USA). A sex-stratified Cox proportional hazard model, using time-on-study as the time scale and adjusted for the continuous covariates of age at the start of follow-up and birth year, was used to compare the risk of leukemia incidence in each occupation and industry group to the rest of the cohort. Follow-up continued until the first occurrence of a leukemia diagnosis, and right censoring was defined by the following endpoints: death, emigration from Ontario, age 85, or end of the study period (31 December 2019). Hazard ratios (HR) were presented at division, major, and select minor levels, along with a corresponding 95% confidence interval (CI). Cells with case counts <6 were not reported due to organizational data privacy policies.

## 3. Results

### 3.1. Cohort Size and Characteristics

The analytic cohort consisted of 2,363,818 workers, of which 35.4% were female. There was a total of 1810 incident leukemia cases among females and 5464 among males that occurred following cohort entry. The median age of cohort entry was 34 years among the entire ODSS cohort and 46 among leukemia cases, with a median follow-up of 23 and 18 years, respectively. Additional cohort characteristics, stratified by sex, are presented in Table 1.

### 3.2. Leukemia Stratified by Sex

Table 2 presents hazard ratios for leukemia and occupation, stratified by sex. At the division level, males in ‘*transport equipment operating*’ occupations were found to have an elevated risk of leukemia, with associated increases at the major level seen in ‘*motor transport operating*’ and ‘*related transport equipment operating*’. Specific minor level occupations with elevated risks included ‘*bus drivers*’ (HR: 1.50, 95% CI 1.12–1.99) and ‘*truck drivers*’ (HR: 1.11, 95% CI 1.02–1.22). Female workers showed similar trends in ‘*transport equipment operating*’ and ‘*motor transport operating*’, with estimates being less statistically precise (i.e., wider confidence intervals). ‘*Protective service*’ occupations were also associated with an elevated risk of leukemia in males, particularly among workers in ‘*firefighting*’ (HR: 1.37, 95% CI 1.06–1.76). When investigating results by industry groups, transportation was associated with a moderate elevation in risk among both sexes (Appendix A).

Male workers in ‘*mining and quarrying*’ occupations showed an elevated risk of leukemia, with associated minor level increases seen among ‘*foremen*’ (HR: 2.29, 95% CI 1.33–3.95) and those in ‘*cutting, handling, and loading*’ occupations (HR: 1.51, 95% CI 1.09–2.07). There were insufficient case counts among female workers to estimate risks associated with these occupation groups. Other major-level occupation groups with elevated risks included ‘*metal machining*’ and ‘*mechanics and repairers*’ among males, with a similar magnitude of increase also observed in female workers. Moreover, both males and females had a moderately elevated risk of leukemia among ‘*metal product fabricating and assembling*’ occupations. When evaluating industry results, ‘*transportation equipment manufacturing*’ showed a moderate elevation in leukemia risk among both sexes, while ‘*furniture and fixture manufacturing*’ was associated with an increase only among female workers. Female workers also had an increased risk among industries involved in ‘*retail trade*’, mirrored by a slight increase seen in ‘*sales*’ occupations.

### 3.3. Leukemia Stratified by Subtype

When exploring associations among each leukemia subtype, low cell counts for ALL, CML, AML, and CLL precluded further stratification by sex. As a result, we presented findings by subtype for both sexes combined only.

As shown in Table 3, workers in ‘*transport equipment operating*’ showed a moderately elevated risk of developing AML, CML, and CLL relative to the rest of the worker cohort. Within this broad occupation group, similar trends were observed among those in ‘*motor transport operating*’. There was a strong elevation in the risk of these same subtypes among workers involved in ‘*related transport equipment operating*’, which consists of foremen, street railway operators, and other transport-related occupations not classified elsewhere. Similar trends were observed in our industry-level analysis, where workers in ‘*transportation*’ industries had an elevated risk of AML, CML, and CLL onset (Appendix A).

Among workers in ‘*mining and quarrying*’ occupations, somewhat elevated risks of AML, CML, and CLL were observed. There were insufficient ALL case counts among workers in mining and quarrying occupations to estimate the risk of this subtype. However, among our analysis by industry classifications, workers in ‘*mines, quarries, and oil wells*’ demonstrated an elevated risk in all four leukemia subtypes.

An increase in the risk of CML, ALL, and CLL was observed among workers in ‘*metal machining*’ occupations, with similar findings among those in ‘*metal product fabrication*’. Workers in ‘*power, lighting and communications: installing and repairing*’ occupations showed a moderately elevated risk in CLL and ALL. In contrast, ‘*other machining and related’* occupations, a category consisting of engravers, buffers, and mold makers, was associated with a strong elevation in AML risk. Lastly, while there was a lower proportion of overall incident ALL cases relative to the other leukemia subtypes in our cohort, a strongly elevated risk was seen for ‘*mechanics and repairing*’ occupations, accompanied by slight increases among other leukemia subtypes. 

For workers employed in ‘*protective service*’ occupations, an elevated risk of CLL was observed relative to the cohort. Moreover, workers in ‘*managerial, administrative and related’* occupations, as well as ‘*other managers and administrators*’ were also associated with a heightened risk of this subtype. An increase in the hazard of CML was seen among ‘*other occupations in medicine and health*’, a major-level category including pharmacists, radiological technicians, and dental hygienists. An increase in this same subtype was also seen among ‘*artistic, literary, recreational and related*’ and ‘*material recording and distributing*’ occupations.

## 4. Discussion

Drawing on data from Ontario’s ODSS worker cohort, this study aimed to identify occupations associated with an increased risk of developing leukemia. Occupation is often used as a surrogate for potential substances and hazards that individuals may be exposed to in their work environments [28]. Overall, we identified several occupation groups associated with elevated risks of leukemia. After adjusting for age at the start of follow-up and birth year, males in protective services, metal machining, transportation, and mining occupations showed a higher incidence of leukemia compared to all other workers in the cohort. These findings appear to be consistent with known carcinogenic hazards in these environments. Similarly, despite being less statistically precise, elevated risks were observed among female workers in transport equipment operating, as well as in metal product machining and fabricating. Beyond smaller case counts among female workers in certain occupations, other factors have been postulated to contribute to differing results, including variations in occupational exposures and the influence of other competing risk factors by sex [23,30]. Dissimilar risks between male and female workers were particularly evident at the industry level, highlighting the possible role of sex as an effect measure modifier, likely driven by differing employment patterns within industries. 

The role of sex as an effect modifier can also be observed in prior cohort studies exploring similar relationships among occupation and industry groups [19,22]. The Nordic Occupational Cancer (NOCCA) Project, representing the largest known cohort study on occupational cancer, followed 15 million workers across Denmark, Finland, Iceland, Norway, and Sweden from 1960 to 2005 [19]. Approaches to create a standardized classification criterion for occupations across these Nordic countries resulted in 53 distinct groups that were used to evaluate cancer risk. Public safety workers, sales agents, and clerical workers were found to have a higher overall incidence of leukemia in comparison to other workers in the cohort. Further analysis by sex revealed an increased incidence of CLL for male farmers and an increased risk of AML for male drivers and sales agents [19]. Among female workers, no significant results for CLL nor AML were observed [19]. Drawing comparisons between this study and our results, analogous risks were observed for drivers. One notable difference between our analysis and this previous study was our use of detailed occupation classifications, which may be better surrogates for carcinogenic occupational exposures.

Our analysis by industry groups revealed an increased risk of leukemia among male and female workers employed in transportation and transport equipment manufacturing. These observations align with our findings among occupation groups, suggesting a higher incidence of leukemia among those working in environments where transportation equipment and other machinery are used. Among female workers in our cohort, industries in furniture and fixture manufacturing, storage, and retail were also associated with increases in leukemia. Furniture and fixture manufacturing may similarly consist of settings with increased exposure to machinery and related hazards. As for retail and storage, these classifications represent a heterogenous group of industries and further investigation may be needed to elucidate the underlying settings that are driving this increased risk for female workers. Contrasting our findings to a cohort study of 11 million Korean workers from 2007–2015, an increased incidence of leukemia was found among industries in machinery manufacturing, as well as waste collection, treatment, and disposal [22]. Within their sex stratified analysis, Lee et al. observed similar results for males, while females experienced an increased risk in land transport, non-metallic mineral product manufacturing, and social work industries [22]. 

The results from our study, and other similar studies, point to associations among settings that involve the use of transportation equipment [19,22]. In particular, the increased risks seen in our study among transport equipment operating, protective services, mining, and quarrying occupations may be partly attributable to hazards originating from combustion engine use. Benzene, formaldehyde, and 1,3-butadiene are all associated with fossil fuels and combustion engine exhaust [31]. These substances are recognized by IARC as Group 1 known carcinogens, with sufficient evidence in humans for their association with leukemia [5]. Higher concentrations of benzene, in particular, are present in gasoline relative to diesel fuel [32]. This compound has also been observed in emissions [33] and overall traffic pollution [34]. Other emission byproducts known as polycyclic aromatic hydrocarbons (PAHs) have been explored for their leukemogenic effects. In a case-control study exploring PAH serum biomarkers, a significant elevation in PAH concentrations was observed among leukemia cases relative to healthy stem-cell donors [35]. Although these specific substances have shown associations with leukemia incidence, results on overall combustion engine exposure have been observed to vary by leukemia subtype. In one Danish study of 1967 leukemia cases and 3381 controls selected from the general population, indicators of traffic-related air pollution were only associated with AML, with a lack of association seen among other leukemias [34]. Smoke from fires contains a similar mix of carcinogens [36].

Broader epidemiological studies exploring the risk of leukemia among transport-related occupations have yielded inconsistent findings. Large cohort studies similar to ours have previously found significant increases among males in driving occupations [19], as well as among females engaged in passenger land transport industries [22]. Another study of 513 leukemia cases and 1087 controls from the United States also found an increased risk among light truck drivers [37]. In contrast, a retrospective cohort study exploring cancer incidence among urban transportation workers using information from the Danish Cancer Registry found no increased risk of leukemia relative to national incidence rates [38]. This retrospective cohort study of 18,174 bus drivers and tramway operators compared the observed incidence among those employed for >3 months to national cancer rates, adjusting for sex, age, and calendar period [38]. In another cohort study among Korean road transportation workers followed from 2002–2015, no increased risk of leukemia was observed when comparing its incidence to both government employees and the remaining worker cohort [39]. One possible limitation of this study was its definition of cases based on hospital admissions [39]. As certain chronic subtypes of leukemia have shown successful management in outpatient settings [40], utilizing data sources from a combination of inpatient and outpatient settings may help with the inclusion of less severe leukemia cases. 

Our study also identified an increased incidence of leukemia among mining and quarrying occupations. Similar to transportation-related occupations, the use of combustion engines or the handling of petroleum products in resource extraction settings may increase the risk of worker exposure to 1,3-butadiene and benzene [33]. Despite these suspected exposures, a prior mining study investigating the impact of diesel exhaust was unable to identify an association with leukemia [41]. Looking more broadly at other occupational studies, no significant increases in leukemia were observed for industries related to mining and quarrying (e.g., mining of coal and lignite, extraction of crude petroleum and natural gas) among a large cohort of Korean workers [22]. However, findings in this study were based on low case counts and wide confidence intervals. While no significant findings in related occupations were observed in a similar large cohort study among Nordic workers, a non-significant positive association was seen for miners and quarry workers (SIR: 1.58; 95% CI: 0.95–2.47) [19]. Various studies have been conducted on petroleum workers, with a systematic review of 22 studies finding a combined mortality and incidence effect size of 1.08 (95% CI: 0.97–1.19) [42]. Further stratification by occupational settings resulted in a significant increase in leukemia among offshore petroleum workers [42]. 

Other increased risks observed among occupations in metal machining, product manufacturing, and assembling within our study suggest a possible relationship to hazards associated with metal work. Within these settings, metal working fluids, degreasers, and cleaning solvents represent possible sources of exposure to known and suspected carcinogenic substances. These substances can include benzene, dichloromethane, trichloroethylene, and mineral oils. Benzene, which has sufficient evidence for an association with AML and limited evidence as a causative agent for CML and CLL [5], has been found in small concentrations among metal cleaning solvents [43]. Dichloromethane, classified as a probable carcinogenic substance by IARC [44], is used as a painting solvent, cleaner, and degreaser [45]. There have also been inconsistent findings among observational studies that specifically investigated its relationship with leukemia [46]. While another cleaning and degreasing solvent, trichloroethylene, has had few observational studies investigating its association with leukemia, an epidemiological summary of existing research did not suggest the presence of any significant association with this outcome [47]. On the other hand, mineral oils have been used as industrial lubricants in manufacturing processes [48], with *untreated/mildly treated* oils classified as known carcinogens by IARC [49]. *Highly treated* mineral oils, which have been increasingly preferred in occupational settings [49,50], have not been classifiable as to their carcinogenic effects in humans [49]. In one investigation of cancer mortality among a cohort of US workers exposed to metal working fluids, no significant associations were found for the exposure to highly refined mineral oils [50]. Among all previously mentioned chemical hazards, it is pertinent to also note that the frequency and level of exposure may have decreased in recent decades. This decrease is most notable for benzene, where greater awareness of its carcinogenic properties led to improved engineering controls and diminishing use in occupational settings [51,52]. As a result, additional considerations related to longitudinal changes in exposure are necessary. Considering the broader epidemiological literature, a national study of Korean workers found significant increases in the risk of leukemia among machinery, engine, and equipment manufacturing industries [22]. 

Among smaller case-control studies, significant increases of AML have been previously identified among workers in the steel industry [21]. In contrast, no significant findings were found for occupations associated with metal work (mechanics and welders) among 15 million workers followed for cancer diagnoses by the NOCCA project [19]. Instead, leukemia rates among these groups were elevated, with standardized incidence ratios of 1.08 (95% CI: 0.92–1.26) and 1.17 (95% CI: 0.70–1.83) for male mechanics and welders, respectively. Given these differing findings, it is pertinent to acknowledge that the heterogeneity seen between different occupational cancer studies may be partly driven by variations in analytic approaches, the extent of exposure to occupational hazards, and the confounding effect of other population characteristics [13,30,53]. 

As discussed in previous investigations using the ODSS [24,54], our study presents several strengths and limitations relative to other occupational cancer studies. First, the use of diagnostic outcomes rather than mortality endpoints reduced the potential for survivorship bias and outcome misclassification. To this effect, we note a previous investigation into the accuracy of death certificates within the United States, where only 65% of underlying malignancies were found to be accurately reported [55]. More specifically, both myeloid and lymphoid leukemias were found to be underreported in death certificates related to cases confirmed by hospital diagnoses [55]. The use of diagnostic information from a provincial cancer registry (OCR) thus presented an opportunity to explore associations using high quality administrative data arising from multiple sources, including hospital admissions, discharge information, pathology, and treatment reports. Similarly, a low risk of occupation and industry misclassification was anticipated due to the use of compensation data (WSIB) which is reviewed and verified during the claims adjudication process. Finally, our use of an internal comparison to calculate our hazard ratios may have reduced the effect of unknown confounders related to healthy worker bias, including lifestyle and socioeconomic factors [56]. Despite these strengths, there are several limitations in our study. Employment information is only captured at a single point in time, lacking comprehensive work history or individual exposure data. As a result, prior employment in occupations different to the one classified for each worker may have confounded our observed associations. This effect may be heightened by the length of prior employment history, particularly for leukemia cases, due to their observed median age of cohort entry being 46 years. The use of administrative data precluded us from obtaining information on potential risk factors, such as smoking status [4]. Moreover, while our use of a cohort of previously injured workers is presumed to be generalizable to all Ontario workers, we may have been unable to effectively capture associations among those working in occupations not requiring mandatory WSIB coverage, such as independent operators. The inability to capture some independent operators, including sole-proprietorship transportation workers and contractors, may have attenuated effect estimates in some of our findings.

## 5. Conclusions

While several occupational exposures are recognized as potential risk factors for leukemia development, significant gaps persist in our understanding of hazards associated with specific occupations. Our study sought to address this gap by identifying occupations and industries associated with an elevated risk of leukemia among a cohort of workers from Ontario (Canada). Relying on a large worker cohort with detailed employment classifications and high-quality cancer incidence data, we found increased risks for males employed in transport operating, metalworking, resource extraction, and protective service occupations. Similar but less statistically precise elevations in risk were observed for female workers in occupations related to transport operating and metalworking. The smaller number of female workers among certain occupations may have contributed to differences in risk estimates and statistical precision seen across male and female workers. Exposure assessments are needed to understand causative factors in occupations linked to leukemia risk, especially by subtype. There is also a need to further explore potential sex differences in the risk of this disease. These findings, in turn, can better inform preventive strategies aimed at reducing the burden of leukemia.

## Figures and Tables

**Table 1 ijerph-21-00981-t001:** Characteristics of leukemia cases and the overall ODSS cohort, 1983 to 2019.

Characteristic	Leukemia Cases		Overall ODSS Cohort	
	Females	Males	Females	Males
N (%) ^a^	1810 (24.9)	5464 (75.1)	836,589 (35.4)	1,527,229 (64.6)
*- AML*	541 (28.7)	1347 (71.4)	-	-
*- CML*	252 (25.2)	747 (74.8)	-	-
*- ALL*	106 (31.8)	227 (68.2)	-	-
*- CLL*	693 (22.2)	2426 (77.8)	-	-
*- Other*	245 (24.4)	759 (75.6)	-	-
Year of Birth, Median (IQR)	1945 (1936–1955)	1944 (1934–1954)	1960 (1950–1970)	1960 (1949–1969)
Age at Cohort Entry, Median (IQR)	47 (38–54)	46 (36–55)	37 (27–48)	33 (25–44)
Age at Diagnosis, Median (IQR) ^b^	63 (55–72)	64 (54–72)	-	-
Years of Follow-Up, Median (IQR) ^c^	17 (9–23)	18 (11–25)	21 (11–29)	25 (15–32)

^a^: Overlapping diagnoses possible among leukemia subtypes; total may exceed 100%; excluding prevalent cases at cohort entry; categorized based on the following SEER recode definitions: AML (35021, 35031), CML (35022), ALL (35011), CLL (35012), and Other (35013, 35023, 35041, 35043). ^b^: Age at diagnosis, excluding prevalent cases at cohort entry. ^c^: Years of follow-up, with losses occurring due to diagnosis, death, emigration, age > 85, or administrative censoring as of 31 December 2019. ODSS: Occupational Disease Surveillance System; AML: acute myeloid leukemia; CML: chronic myeloid leukemia; ALL: acute lymphocytic leukemia; CLL: chronic lymphocytic leukemia; Other: other leukemia not otherwise classified; IQR: interquartile range.

**Table 2 ijerph-21-00981-t002:** Hazard ratios and 95% confidence intervals for leukemia by division and major level occupation groups, stratified by sex, ODSS, 1983–2019.

Occupation Group (CCDO Code)	Leukemia ^a^			
	Females (n = 1810 Cases)		Males (n = 5464 Cases)	
	Cases (Workers) ^b^	HR (95% CI) ^c^	Cases (Workers) ^b^	HR (95% CI) ^c^
Managerial, Administrative and Related (11)	40 (21,015)	1.14 (0.83–1.56)	68 (16,786)	1.18 (0.93–1.50)
*- Officials and Administrators, Government (111)*	8 (3719)	1.25 (0.62–2.50)	21 (3149)	1.50 (0.98–2.31)
*- Managers and Administrators—Other (113/4)*	21 (7881)	1.47 (0.95–2.26)	32 (8833)	1.11 (0.78–1.57)
*- Related to Management and Administration (117)*	14 (9699)	0.94 (0.56–1.60)	16 (5013)	1.03 (0.63–1.69)
Natural Sciences, Engineering and Mathematics (21)	11 (6336)	1.07 (0.59–1.93)	72 (22,734)	0.97 (0.77–1.22)
*- Physical Sciences (211)*	<6 (770)	-	6 (2588)	0.58 (0.26–1.29)
*- Life Sciences (213)*	<6 (893)	-	9 (1997)	1.45 (0.76–2.79)
*- Architects and Engineers (214/5)*	<6 (821)	-	17 (4030)	1.03 (0.64–1.65)
*- Architecture and Engineering—Other (216)*	6 (2579)	1.57 (0.70–3.50)	41 (13,034)	1.03 (0.76–1.41)
Social Sciences and Related Fields (23)	45 (28,038)	1.09 (0.81–1.47)	22 (7808)	1.14 (0.75–1.73)
*- Social Work and Related Fields (233)*	34 (21,811)	1.06 (0.76–1.49)	16 (5858)	1.10 (0.67–1.80)
*- Social Sciences and Related Fields—Other (239)*	7 (3692)	1.69 (0.80–3.55)	<6 (959)	-
Teaching and Related (27)	94 (49,487)	1.03 (0.84–1.27)	48 (11,739)	1.15 (0.87–1.53)
*- Elementary/Secondary School Teaching and Related (273)*	83 (45,358)	1.01 (0.81–1.26)	36 (9762)	1.07 (0.77–1.48)
*- Teaching and Related—Other (279)*	11 (3488)	1.44 (0.79–2.60)	9 (1679)	1.36 (0.70–2.61)
Medicine and Health (31)	283 (136,218)	1.01 (0.89–1.15)	54 (19,740)	0.99 (0.76–1.29)
*- Nursing Therapy and Related Assisting (313)*	264 (123,613)	1.01 (0.88–1.15)	45 (16,950)	0.97 (0.72–1.30)
*- Medicine and Health—Other (315)*	23 (15,126)	1.15 (0.76–1.74)	11 (2912)	1.42 (0.78–2.56)
Art, Literary, Recreational and Related (33)	10 (8423)	1.16 (0.62–2.16)	24 (9780)	1.04 (0.69–1.55)
*- Fine and Commercial Art Photography (331)*	<6 (1313)	-	6 (1709)	1.17 (0.53–2.60)
*- Performing and Audiovisual Arts (333)*	<6 (888)	-	6 (1983)	1.47 (0.66–3.27)
*- Sport and Recreation (337)*	<6 (5775)	-	11 (5845)	0.84 (0.47–1.53)
Clerical and Related (41)	241 (110,024)	1.00 (0.88–1.15)	327 (102,568)	1.02 (0.91–1.14)
*- Stenographic and Typing (411)*	35 (10,860)	1.03 (0.74–1.44)	<6 (309)	-
*- Bookkeepers, Tellers and Cashiers, and Other Clerks (413)*	62 (33,175)	0.95 (0.74–1.23)	16 (4801)	1.27 (0.78–2.07)
*- Office Machine and Electronic Data Processing Operators (414)*	7 (3476)	0.77 (0.37–1.63)	<6 (1012)	-
*- Material Recording Scheduling and Distributing (415)*	35 (15,344)	1.06 (0.76–1.48)	196 (61,401)	1.04 (0.90–1.20)
*- Reception, Information, Mail and Message Distribution (417)*	49 (18,744)	1.18 (0.89–1.57)	86 (25,072)	1.00 (0.81–1.23)
*- Clerical and Related—Other (419)*	68 (30,558)	1.14 (0.90–1.46)	29 (12,030)	0.84 (0.58–1.21)
Sales (51)	170 (86,051)	1.14 (0.98–1.34)	183 (78,594)	0.99 (0.85–1.14)
*- Sales Occupations in Commodities (513/4)*	167 (82,357)	1.17 (0.99–1.37)	158 (73,137)	0.94 (0.80–1.10)
*- Other Sales Occupations (519)*	<6 (3945)	-	24 (5505)	1.43 (0.96–2.14)
Services (61)	408 (205,550)	0.94 (0.84–1.05)	656 (203,809)	1.02 (0.94–1.11)
*- Protective Services (611)*	15 (13,222)	0.86 (0.52–1.42)	198 (53,608)	1.17 (1.02–1.35)
*- Food And Beverage Preparation (612)*	171 (90,984)	1.05 (0.89–1.22)	103 (57,186)	0.92 (0.75–1.11)
*- Lodging and Other Accommodation (613)*	22 (6270)	0.97 (0.63–1.47)	<6 (2866)	-
*- Personal Services (614)*	82 (42,029)	0.95 (0.76–1.19)	20 (4854)	1.44 (0.93–2.23)
*- Apparel and Furnishing Services (616)*	16 (5907)	0.91 (0.56–1.49)	11 (2600)	1.21 (0.67–2.19)
*- Other Services (619)*	137 (58,846)	0.88 (0.74–1.05)	331 (88,811)	0.96 (0.86–1.07)
Farming, Horticulture and Animal Husbandry (71)	18 (12,576)	0.97 (0.61–1.54)	90 (42,962)	0.81 (0.66–1.00)
*- Other (718/9)*	17 (12,241)	0.95 (0.59–1.53)	86 (41,609)	0.82 (0.66–1.01)
Forestry and Logging (75)	<6 (648)	-	38 (10,327)	0.96 (0.69–1.31)
Mining and Quarrying, Including Oil and Gas (77)	<6 (196)	-	78 (13,349)	1.28 (1.02–1.60)
Processing: Mineral, Metal, Chemical (81)	38 (17,175)	1.02 (0.74–1.40)	211 (65,357)	0.95 (0.83–1.09)
*- Mineral Ore Treating (811)*	<6 (37)	-	7 (943)	1.42 (0.68–2.99)
*- Metal Processing and Related (813/4)*	<6 (2181)	-	98 (27,195)	0.98 (0.80–1.19)
*- Clay Glass and Stone Forming and Related (815)*	<6 (1187)	-	44 (9301)	1.27 (0.94–1.71)
*- Chemicals Petroleum Rubber Plastic and Related (816/7)*	30 (13,930)	1.01 (0.70–1.45)	70 (29,454)	0.81 (0.64–1.03)
Processing: Food, Wood, Textile (82)	67 (34,321)	0.89 (0.70–1.13)	240 (70,404)	1.05 (0.92–1.20)
*- Food and Beverage and Related Processing (821/2)*	52 (26,592)	0.91 (0.69–1.20)	182 (47,255)	1.15 (1.00–1.34)
*- Wood Processing Occupations Except Paper Pulp (823)*	<6 (779)	-	16 (6167)	0.83 (0.51–1.36)
*- Pulp and Papermaking and Related Occupations (825)*	<6 (854)	-	24 (4984)	1.20 (0.80–1.79)
*- Textile Processing (826/7)*	7 (4517)	0.60 (0.29–1.26)	12 (5461)	0.63 (0.36–1.11)
*- Other Processing (829)*	<6 (1824)	-	14 (7609)	0.94 (0.56–1.60)
Machining and Related (83)	51 (22,255)	0.97 (0.73–1.28)	667 (173,587)	1.04 (0.95–1.12)
*- Metal Machining (831)*	7 (2771)	1.17 (0.56–2.46)	201 (40,575)	1.23 (1.07–1.41)
*- Metal Shaping and Forming, Except Machining (833)*	39 (17,304)	0.94 (0.69–1.29)	433 (122,099)	0.98 (0.89–1.08)
*- Wood Machining (835)*	<6 (1227)	-	27 (7744)	1.09 (0.75–1.60)
*- Clay, Glass, Stone, and Related Machining (837)*	<6 (143)	-	6 (1593)	0.88 (0.40–1.96)
*- Machining and Related—Other (839)*	<6 (1480)	-	50 (11,389)	1.08 (0.82–1.42)
Product Fabricating, Assembling and Repairing (85)	195 (70,353)	1.08 (0.93–1.25)	1037 (274,503)	1.03 (0.96–1.10)
*- Fabricating and Assembling, Metal Products (851/2)*	59 (20,867)	1.29 (0.99–1.67)	298 (69,482)	1.10 (0.98–1.24)
*- Electrical and Electronic and Related Equipment (853)*	31 (12,584)	0.85 (0.59–1.21)	88 (27,897)	0.93 (0.76–1.15)
*- Wood Products (854)*	12 (3053)	1.64 (0.93–2.89)	53 (22,313)	0.74 (0.57–0.97)
*- Textile Fur and Leather Products (855/6)*	70 (18,206)	1.16 (0.91–1.47)	39 (8745)	1.06 (0.77–1.45)
*- Rubber Plastic and Related Products (857)*	8 (4245)	0.69 (0.34–1.38)	40 (11,056)	1.00 (0.73–1.37)
*- Mechanics and Repairers Except Electrical (858)*	6 (2985)	1.20 (0.54–2.67)	490 (116,723)	1.10 (1.00–1.20)
*- Product Fabricating Assembling and Repairing—Other (859)*	24 (12,756)	0.91 (0.61–1.37)	119 (37,838)	1.05 (0.87–1.26)
Construction Trades (87)	12 (5987)	1.50 (0.85–2.64)	837 (227,559)	1.06 (0.98–1.14)
*- Excavating, Grading, Paving and Related (871)*	<6 (519)	-	79 (19,078)	0.98 (0.78–1.22)
*- Electrical Power Lighting and Wire Communications (873)*	<6 (1355)	-	142 (37,100)	1.04 (0.88–1.23)
*- Construction Trades Occupations—Other (878/9)*	8 (4144)	1.66 (0.83–3.32)	630 (175,199)	1.07 (0.98–1.16)
Transport Equipment Operating (91)	31 (16,842)	1.10 (0.77–1.57)	663 (166,284)	1.15 (1.06–1.25)
*- Air Transport Operating (911)*	<6 (1586)	-	17 (8255)	0.92 (0.57–1.49)
*- Railway Transport Operating (913)*	<6 (239)	-	18 (4098)	1.04 (0.65–1.65)
*- Water Transport Operating (915)*	<6 (198)	-	8 (2657)	0.73 (0.37–1.47)
*- Motor Transport Operating (917)*	24 (9937)	1.21 (0.81–1.81)	586 (141,619)	1.14 (1.05–1.25)
*- Transport and Related Equipment Operating—Other (919)*	7 (5764)	0.96 (0.45–2.02)	58 (14,971)	1.55 (1.19–2.02)
Materials Handling and Related (93)	89 (33,556)	1.08 (0.88–1.34)	398 (128,621)	1.02 (0.92–1.13)
Other Crafts and Equipment Operating (95)	16 (7000)	0.87 (0.53–1.43)	97 (22,348)	1.10 (0.90–1.35)
*- Printing and Related (951)*	13 (6076)	0.78 (0.45–1.35)	51 (13,858)	1.04 (0.79–1.37)
*- Stationary Engine and Utilities Equipment Operating (953)*	<6 (420)	-	44 (7603)	1.22 (0.90–1.64)

^a^: Leukemia (excluding prevalent cases at cohort entry) is defined by the earliest date of diagnosis for any of the following SEER recode definitions: 35021, 35031, 35022, 35011, 35012, 35013, 35023, 35041, 35043. ^b^: Occupation groups with case counts <6 for both sexes suppressed. ^c^: Adjusted for age at start of follow-up and birth year. CCDO, Canadian Classification Dictionary of Occupation; ODSS, Occupational Disease Surveillance System; HR, hazard ratio; CI: confidence interval.

**Table 3 ijerph-21-00981-t003:** Hazard ratios and 95% confidence intervals for leukemia subtypes, by division and major level occupational groups, ODSS, 1983–2019.

Occupation Group (CCDO Code)	Total Workers ^b^	Acute Myeloid (n = 1888) ^a^		Chronic Myeloid (n = 999) ^a^		Acute Lymphocytic (n = 333) ^a^		Chronic Lymphocytic (n = 3119) ^a^	
		Cases	HR (95% CI) ^c^	Cases	HR (95% CI) ^c^	Cases	HR (95% CI) ^c^	Cases	HR (95% CI) ^c^
Managerial, Administrative and Related (11)	37,801	24	0.94 (0.63–1.40)	9	0.71 (0.37–1.38)	<6	-	54	1.38 (1.05–1.81)
*- Officials and Administrators, Government (111)*	6868	9	1.65 (0.86–3.18)	<6	-	<6	-	14	1.59 (0.94–2.69)
*- Managers and Administrators—Other (113/4)*	16,714	8	0.67 (0.34–1.35)	6	1.04 (0.46–2.32)	<6	-	29	1.57 (1.09–2.27)
*- Related to Management and Administration (117)*	14,712	7	0.81 (0.39–1.71)	<6	-	<6	-	14	1.13 (0.67–1.91)
Natural Sciences, Engineering and Mathematics (21)	29,070	28	1.29 (0.89–1.87)	7	0.60 (0.29–1.27)	<6	-	40	1.09 (0.80–1.49)
*- Life Sciences (211)*	2890	<6	-	<6	-	<6	-	6	1.84 (0.82–4.09)
*- Architects and Engineers (214/5)*	4851	<6	-	<6	-	<6	-	11	1.37 (0.76–2.48)
*- Architecture and Engineering—Other (216)*	15,613	19	1.72 (1.09–2.70)	<6	-	<6	-	20	1.06 (0.68–1.65)
Social Sciences and Related Fields (23)	35,846	21	1.21 (0.79–1.87)	7	0.81 (0.38–1.70)	<6	-	25	1.07 (0.72–1.58)
*- Social Work and Related Fields (233)*	27,669	14	1.06 (0.62–1.79)	<6	-	<6	-	20	1.12 (0.72–1.74)
Teaching and Related (27)	61,226	40	1.05 (0.76–1.44)	11	0.59 (0.33–1.08)	7	1.03 (0.48–2.20)	66	1.25 (0.97–1.59)
*- Elementary/Secondary School Teaching and Related (273)*	55,120	33	0.98 (0.69–1.39)	9	0.55 (0.29–1.07)	<6	-	57	1.23 (0.95–1.61)
*- Teaching and Related Occupations—Other (279)*	5167	<6	-	<6	-	<6	-	10	1.68 (0.90–3.13)
Medicine and Health (31)	155,958	90	0.91 (0.73–1.13)	52	1.14 (0.85–1.53)	22	1.19 (0.75–1.87)	132	1.01 (0.84–1.21)
*- Nursing Therapy and Related Assisting (313)*	140,563	86	0.95 (0.75–1.19)	45	1.06 (0.77–1.45)	21	1.23 (0.77–1.97)	119	0.98 (0.81–1.19)
*- Medicine and Health—Other (315)*	18,038	6	0.73 (0.33–1.63)	9	2.32 (1.20–4.49)	<6	-	14	1.29 (0.76–2.19)
Artistic, Literary, Recreational and Related (33)	18,203	7	0.83 (0.39–1.73)	10	2.24 (1.20–4.19)	<6	-	8	0.61 (0.30–1.21)
Clerical and Related (41)	212,592	143	0.94 (0.79–1.11)	93	1.23 (0.99–1.52)	28	0.98 (0.66–1.45)	239	1.03 (0.90–1.18)
*- Stenographic and Typing (411)*	11,169	8	0.75 (0.38–1.52)	8	1.71 (0.85–3.46)	<6	-	16	1.14 (0.69–1.87)
*- Bookkeepers, Tellers and Cashiers, and Other Clerks (413)*	37,976	21	0.93 (0.60–1.43)	9	0.82 (0.43–1.60)	<6	-	37	1.24 (0.89–1.72)
*- Material Recording Scheduling and Distributing (415)*	76,745	56	0.99 (0.76–1.29)	44	1.45 (1.07–1.97)	<6	-	97	1.02 (0.83–1.25)
*- Reception, Information, and Mail (417)*	43,816	32	0.94 (0.66–1.33)	23	1.32 (0.87–2.00)	7	1.17 (0.55–2.47)	54	1.00 (0.76–1.31)
*- Clerical and Related—Other (419)*	42,588	29	1.08 (0.75–1.56)	8	0.61 (0.30–1.23)	9	1.84 (0.94–3.61)	42	1.11 (0.82–1.51)
Sales (51)	164,645	98	1.07 (0.87–1.31)	42	0.89 (0.65–1.21)	18	0.98 (0.61–1.58)	147	1.07 (0.91–1.27)
*- Sales Occupations in Commodities (513/4)*	155,494	91	1.06 (0.86–1.31)	37	0.83 (0.60–1.16)	16	0.92 (0.55–1.53)	135	1.06 (0.89–1.26)
*- Sales Occupations in Services (517)*	1140	<6	-	<6	-	<6	-	<6	-
*- Other Sales Occupations (519)*	9450	7	1.22 (0.58–2.57)	<6	-	<6	-	13	1.42 (0.82–2.44)
Services (61)	409,359	295	1.03 (0.91–1.17)	159	1.12 (0.94–1.33)	49	0.93 (0.68–1.26)	409	0.89 (0.80–0.98)
*- Protective Services (611)*	66,830	46	0.96 (0.72–1.29)	24	0.93 (0.62–1.40)	7	0.82 (0.39–1.75)	104	1.29 (1.06–1.57)
*- Food And Beverage Preparation (612)*	148,170	86	1.13 (0.90–1.40)	45	1.16 (0.86–1.58)	20	1.22 (0.77–1.93)	87	0.78 (0.63–0.97)
*- Lodging and Other Accommodation (613)*	9136	8	0.85 (0.42–1.71)	<6	-	<6	-	8	0.57 (0.28–1.13)
*- Personal Services (614)*	46,883	36	1.23 (0.88–1.72)	20	1.48 (0.94–2.33)	<6	-	30	0.76 (0.52–1.09)
*- Apparel and Furnishing Services (616)*	8507	9	1.19 (0.62–2.30)	<6	-	<6	-	14	1.27 (0.75–2.15)
*- Other Services (619)*	147,657	124	0.94 (0.78–1.13)	67	1.02 (0.80–1.31)	20	0.95 (0.60–1.49)	184	0.84 (0.72–0.97)
Farming, Horticulture and Animal Husbandry (71)	55,538	21	0.63 (0.41–0.97)	17	0.93 (0.58–1.51)	6	0.90 (0.40–2.03)	50	0.91 (0.68–1.20)
*- Farming, Horticultural and Husbandry—Other (718/9)*	53,850	20	0.63 (0.41–0.98)	17	0.98 (0.61–1.59)	6	0.94 (0.42–2.11)	48	0.92 (0.69–1.22)
Forestry and Logging (75)	10,975	9	0.90 (0.47–1.74)	<6	-	<6	-	17	0.95 (0.59–1.53)
Mining and Quarrying, Including Oil and Gas (77)	13,545	17	1.13 (0.70–1.82)	11	1.37 (0.75–2.48)	<6	-	38	1.37 (0.99–1.89)
Processing: Mineral, Metal, Chemical (81)	82,532	60	0.91 (0.70–1.17)	44	1.25 (0.92–1.69)	7	0.57 (0.27–1.20)	101	0.90 (0.74–1.10)
*- Metal Processing and Related (813/4)*	29,376	23	0.87 (0.58–1.32	19	1.35 (0.85–2.12)	<6	-	38	0.82 (0.59–1.12)
*- Clay Glass and Stone Forming and Related (815)*	10,488	13	1.38 (0.80–2.39)	9	1.74 (0.90–3.36)	<6	-	18	1.09 (0.68–1.73)
*- Chemicals Petroleum Rubber Plastic and Related (816/7)*	43,384	24	0.79 (0.53–1.18)	17	1.05 (0.65–1.69)	<6	-	46	0.95 (0.71–1.28)
Processing: Food, Wood, Textile (82)	104,725	68	0.85 (0.67–1.09)	41	0.98 (0.72–1.34)	21	1.46 (0.94–2.27)	136	1.05 (0.89–1.25)
*- Food and Beverage and Related Processing (821/2)*	73,847	52	0.92 (0.70–1.22)	31	1.05 (0.73–1.50)	16	1.58 (0.95–2.61)	103	1.13 (0.93–1.37)
*- Wood Processing Except Paper Pulp (823)*	6946	<6	-	<6	-	<6	-	8	0.89 (0.44–1.78)
*- Pulp and Papermaking and Related (825)*	5838	<6	-	<6	-	<6	-	14	1.44 (0.85–2.43)
*- Textile Processing (826/7)*	9978	<6	-	<6	-	<6	-	7	0.54 (0.26–1.14)
*- Other Processing (829)*	9433	<6	-	<6	-	<6	-	7	0.93 (0.44–1.95)
Machining and Related (83)	195,842	176	1.00 (0.86–1.17)	108	1.16 (0.94–1.42)	28	0.91 (0.62–1.35)	330	1.08 (0.97–1.22)
*- Metal Machining (831)*	43,346	40	0.94 (0.68–1.28)	36	1.61 (1.15–2.25)	8	1.17 (0.58–2.37)	104	1.38 (1.13–1.68)
*- Metal Shaping and Forming, Except Machining (833)*	139,403	122	1.00 (0.83–1.21)	70	1.07 (0.83–1.36)	20	0.93 (0.59–1.47)	210	0.99 (0.86–1.14)
*- Wood Machining (835)*	8971	7	1.00 (0.48–2.10)	<6	-	<6	-	15	1.26 (0.76–2.10)
*- Machining and Related—Other (839)*	12,869	20	1.61 (1.03–2.50)	8	1.18 (0.59–2.38)	<6	-	20	0.90 (0.58–1.41)
Product Fabricating, Assembling and Repairing (85)	344,856	312	1.03 (0.91–1.16)	180	1.14 (0.97–1.34)	63	1.23 (0.94–1.63)	526	1.01 (0.92–1.11)
*- Fabricating and Assembling, Metal Products (851/2)*	90,349	75	0.93 (0.73–1.17)	55	1.29 (0.98–1.69)	18	1.28 (0.80–2.07)	165	1.21 (1.03–1.42)
*- Electrical and Electronic and Related Equipment (853)*	40,481	37	1.08 (0.78–1.49)	15	0.84 (0.51–1.40)	8	1.35 (0.67–2.73)	50	0.89 (0.67–1.18)
*- Wood Products (854)*	25,366	24	1.22(0.81–1.82)	9	0.82 (0.42–1.58)	<6	-	20	0.59 (0.38–0.91)
*- Textile Fur and Leather Products (855/6)*	26,951	31	1.15 (0.80–1.64)	18	1.37 (0.86–2.19)	<6	-	48	1.19 (0.89–1.59)
*- Rubber Plastic and Related Products (857)*	15,301	8	0.61 (0.30–1.21)	<6	-	<6	-	21	0.97 (0.63–1.49)
*- Mechanics and Repairers Except Electrical (858)*	119,708	119	1.06 (0.88–1.28)	65	1.07 (0.83–1.38)	30	1.73 (1.18–2.54)	220	1.09 (0.95–1.25)
*- Fabricating Assembling and Repairing—Other (859)*	50,594	37	1.01 (0.73–1.41)	25	1.30 (0.87–1.94)	<6	-	61	1.02 (0.79–1.32)
Construction Trades (87)	233,546	219	1.11 (0.96–1.29)	88	0.78 (0.63–0.98)	34	1.02 (0.71–1.47)	388	1.10 (0.98–1.22)
*- Excavating, Grading, Paving and Related (871)*	19,597	17	0.84 (0.52–1.35)	<6	-	<6	-	39	1.06 (0.77–1.45)
*- Power, Lighting and Communications (873)*	38,455	35	1.01 (0.72–1.42)	11	0.58 (0.32–1.05)	9	1.59 (0.82–3.11)	82	1.34 (1.08–1.68)
*- Construction Trades—Other (878/9)*	179,343	170	1.16 (0.99–1.37)	73	0.89 (0.70–1.13)	25	1.00 (0.66–1.51)	275	1.04 (0.91–1.18)
Transport Equipment Operating (91)	183,126	185	1.22 (1.05–1.43)	97	1.19 (0.96–1.47)	20	0.75 (0.48–1.19)	310	1.17 (1.04–1.32)
*- Air Transport Operating (911)*	9841	<6	-	<6	-	<6	-	6	0.69 (0.31–1.54)
*- Motor Transport Operating (917)*	151,556	161	1.22 (1.03–1.43)	90	1.26 (1.01–1.57)	17	0.74 (0.45–1.21)	271	1.16 (1.02–1.31)
*- Transport and Related Operating—Other (919)*	20,735	19	1.57 (1.00–2.48)	11	1.79 (0.98–3.27)	<6	-	30	1.58 (1.10–2.28)
Materials Handling and Related (93)	162,177	100	0.81 (0.66–0.99)	71	1.08 (0.85–1.38)	16	0.68 (0.41–1.12)	223	1.11 (0.97–1.27)
Other Crafts and Equipment Operating (95)	29,348	31	1.14 (0.80–1.63)	12	0.83 (0.47–1.47)	6	1.30 (0.58–2.92)	52	1.12 (0.85–1.47)
*- Printing and Related (951)*	19,934	17	1.00 (0.62–1.61)	<6	-	<6	-	30	1.08 (0.75–1.54)
*- Stationary Engine and Utilities Equipment (953)*	8023	13	1.42 (0.82–2.46)	7	1.47 (0.70–3.09)	<6	-	21	1.25 (0.81–1.92)

^a^: Cases defined by the earliest date of diagnosis for each leukemia subtype, based on the following SEER recode definitions: Acute Myeloid (35021, 35031), Chronic Myeloid (35022), Acute Lymphocytic (35011), Chronic Lymphocytic (35012); excluding prevalent cases at cohort entry. ^b^: Occupation groups with case counts <6 for all subtypes suppressed. ^c^: Adjusted for age at start of follow-up, birth year, and sex. CCDO, Canadian Classification Dictionary of Occupation; ODSS, Occupational Disease Surveillance System; HR, hazard ratio; CI: confidence interval.

## Data Availability

Supporting data is available upon request and can be accessed if conditions are met that comply with organization guidelines.

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
