# Peer review of "Leukemia Incidence by Occupation and Industry: A Cohort Study of 2.3 Million Workers from Ontario, Canada"

_ijerph, 2024, doi:10.3390/ijerph21080981_

Round 1
Reviewer 1 Report
Comments and Suggestions for Authors
Author Response
We thank the reviewer for their comments and suggestions.
We have copied each comment below and added a description of the specific revisions made to the paper. In addition, we have used the track changes feature to highlight where changes have been made to the original submission (see attached word document).
Comment: [Abstract, Lines 8-22] Add in the abstract when the surveillance started
Response: In response to the reviewer’s comment, we have included a brief statement in the abstract clarifying the earliest date of cohort entry. Workers with WSIB claims between 1983 and 2019 were eligible for inclusion in the ODSS cohort [Abstract, Page 1].
Comment: [Abstract, Lines 8-22] “Analysis underscores the need for exposure-based occupational studies to better understand suspected drivers of leukemia in these settings.” I do not completely follow this…..might refer to the specific jobs found needing an exposure assessment approach
Response: We thank the reviewer for their comment, we have revised the last sentence of the abstract to clarify our position on this statement. The reviewer is correct in assuming that this refers to the need to conduct exposure assessments within the specific employment groups highlighted by our analysis [Abstract, Page 1].
Overall, while we rely on previous literature to highlight possible occupational hazards that may help explain the heightened risk of leukemia in these employment settings, we are unable to provide definitive assertions regarding specific exposures that contribute to these increased risks. In this last sentence, we highlight the need for exposure-based studies to better understand their contribution to the development of leukemia in the highlighted occupations and industries.
Comment: [Methods, Line 102] What does this mean: “Lost-time compensation claims data was used to define each worker’s occupation and industry.” Wasn’t this available independent of claims data?
Response: We appreciate this question by the reviewer. We have revised this section in response to another reviewer’s comment and believe these previous changes may help clarify the method of obtaining information on each worker’s occupation and industry [Methods, Section 2.2, Page 3]. Each worker’s occupation and industry was reported at the time of a lost time compensation claim and was coded by WSIB based on the 1971 Canadian Classification Dictionary of Occupation (CCDO) and the Standard Industry Classification (SIC) systems.
Comment: Can the analysis address those with the same jobs for 20+ years? Consider jobs in a given industry, such as transportation.
Response: Due to the nature of the administrative data used in this study, we do not have longitudinal data on each worker’s employment. In response to another reviewer’s comment, we have included an additional commentary on this matter in our limitations section [Discussion, Page 12]
Comment: How good is the subtyping of leukemia by the various data sources? Could this influence the results?
Response: Our cancer incidence data was obtained from the Ontario Cancer Registry (OCR). The OCR is continually updated and collects information pertaining to the diagnosis of primary cancers across Ontario, Canada. The OCR conforms to standards set by the National Cancer Institute’s SEER program, and integrates data from multiple sources, including hospital admission and discharge records, pathology reports from hospitals and community-level sources, as well as consultation and treatment records of patients referred to regional cancer centers. The use of multiple data sources helps the OCR achieve a high level of accuracy. Due to these factors, we believe the risk of misclassification bias within our cancer incidence data is minimal.
Comment: Add a table of selected industry results in the main paper as this important for the results.
Response: We thank the reviewer for their suggestion. While we are amenable with moving our industry results tables from the supplementary section to the main text if needed, we believe our current approach effectively highlights key findings at the industry level. More specifically, we believe that discussing several industry results in the main text and referring readers to the supplementary tables helps ensure that our manuscript is both informative and concise. In our manuscript, we made an effort to place a greater emphasis on our findings among occupation groups. We briefly discuss this choice in our methods section [Section 2.2, Page 3], stating “While industry results are reported, a greater emphasis was placed on occupation groups, as it is better regarded as a means to represent occupational hazards and other workplace characteristics”.

Reviewer 2 Report
Comments and Suggestions for Authors
the article is very topical and within the scope of the newspaper.
it just needs a few details to be improved in the methodology and conclusions section

The English is correct. just need a minor corretions
Author Response
We thank the reviewer for their comments and suggestions.
We have copied each comment below and added a description of the specific revisions made to the paper. In addition, we have used the track changes feature to highlight where changes have been made to the original submission (see attached word document).
Comment: [Lines 8-22] The reviewer suggested formatting the abstract to mirror the structure of the main text.
Response: We thank the reviewer for their feedback. We believe that the current abstract follows the style and formatting guidelines recommended by the journal. Pertinent information related to the background, methods, results, and conclusions arising from our study are presented in a structured format, albeit without headings. For these reasons, we did not revise the structure of the abstract.
Comment: [Line 23] The reviewer suggested revising the keywords of the article so that they are different from the title. The reviewer further noted a repetition of terms in the keywords section.
Response: In response to the reviewer’s comment, we revised the keywords section to reduce the repetition of some terminology (including ‘occupation’ and ‘leukemia’). A new keyword discussing the study design was added [Page 1].
Comment: [Lines 33-36] The reviewer asked the authors to specify the previous literature that supports our statements on occupational exposures.
Response: We made a slight revision to the sentence to clarify that the other literature complementing the IARC evidence were reviews of previous observational studies in this domain [Introduction, Page 1].
Comment: [Lines 77-79] The reviewer suggested that the authors be more specific with their study objectives.
Response: We revised wording of this sentence to clarify that this surveillance study is aiming to identify occupation and industry groups associated with an elevated risk of leukemia [Introduction, Page 2].
Comment: [Lines 80-134] The reviewer suggested revising the methods section so that it can be reproduced by other authors. Moreover, the reviewer noted that there are some details missing with regards to the ‘professional activities’ of our study population.
Response: We thank the reviewer for their comment. At present, we believe that the methods section contains a sufficient level of detail to facilitate the reproducibility of our study. In this section, we cite a previous paper that described the data linkage process for the ODSS. Additional details are provided regarding our selection criteria and our multivariate analysis. Regarding the classification of ‘professional activities’, which we assume refers to the employment and industry classification for each worker, we believe that reporting the occupation and industry classification systems is sufficient for replication as these classification systems are widely available. Due to these reasons, we believe no edits are needed in response to this comment.
Comment: [Concluding paragraph, Page 13] The reviewer pointed to the following questions at the manuscript’s concluding paragraph: “Why is your work scientifically relevant?”, “What is new about it compared to existing work?”, “What are the main advantages of your work?”, and “What are the main difficulties?”
Response: Regarding the scientific relevance, we state that gaps remain in our understanding of occupational hazards associated with an elevated risk of leukemia. The main limitations of our work are reported in the discussion section, and we believe that there would be limited value in repeating this information in the conclusion. However, we have added a brief statement regarding strengths of our study and novelty compared to prior research was added to the concluding paragraph [Conclusion, Page 13].

Reviewer 3 Report
Comments and Suggestions for Authors
Overall a well-written, informative manuscript that will be of interest to a range of readers. I have a few specific comments and questions, attached.

Author Response
We thank the reviewer for their comments and suggestions.
We have copied each comment below and added a description of the specific revisions made to the paper. In addition, we have used the track changes feature to highlight where changes have been made to the original submission (see attached word document).
Comment: Lines 73-75 state “While understanding the role of sex as an effect measure modifier requires further investigation, it has been suggested that physiological differences and varying patterns of exposure to occupational hazards may be contributing factors [23].” Here it would be good to mention a little more detail on what ref 23 says, is the point that within a given industry or occupation, there is reason to believe that women have different (less?) opportunity for chemical exposure than men do? Just mention what the authors of ref 23 said in a little more detail.
Response: We thank the reviewer for this suggestion. We note that the reference (ref 23) specifies multiple hypotheses that may help explain differences in occupational cancer risk between male and female workers. We have revised this section to outline factors that could contribute to differences in occupational hazard exposures, namely frequency and intensity [Introduction, Page 2]. We are unable to make a generalized statement that female workers have less of an opportunity for chemical exposure, as it may vary based on the specific occupation and a number of other contextual factors.
Comment: Line 87 says that lost time claims as early as 1983 were entered into the cohort, with a median age of cohort entry of 47 yrs for females, 46 years for males. So if people started working at age 20, they could have at least 25 years of employment prior to cohort entry. This raises the possibility (likelihood) that cohort members could have had prior employment, and exposures different than those reported at the time they entered the cohort. Also the people with the earliest cohort entry date (1983) could have started working in the 1960s if they had about 25 years employment before cohort entry. The point is that some cohort members were working in years when exposure levels to chemicals like benzene and formaldehyde were much higher than in later years (say 1980). This doesn’t invalidate the study findings, but it should be mentioned in the discussion.
Response: We thank the reviewer for their comment. We revised the limitations section to discuss how prior employment in occupations different to the one classified at cohort entry may have confounded our findings. We further noted that this effect may be heightened by the length of prior employment history for those with leukemia, given their median age of cohort entry was 46 [Discussion, Page 12].
Comment: Lines 88-89 say that “Accepted lost time claims from 1983 to 2019 (n=2,387,756) were obtained from Ontario’s Workplace Safety Insurance Board (WSIB), with records including the sex, birth date, occupation, and industry classification for each claimant.” Can the authors include a little more info on what is in these records? Is the occupation and industry info limited to what is reported at the date of the claim (so there is no information on longest or usual occupation or industry?) If that is the case, it should be discussed more extensively in the study limitations.
Response: The study is limited to employment information contained at the time of a lost time compensation claim’s submission. Resultingly, we do not have information on the complete employment histories of workers nor information on their longest held occupation. We revised section 2.2 to clarify that we classified occupation and industry at the time of claim [Methods, Page 3]. We did not make further revisions to the limitations section. However, changes made to the limitations section in response to the previous comment may help highlight our lack of employment history information for each worker.
Comment: Lines 90-92 say that “Overall, approximately three quarters of Ontario’s workforce receives WSIB coverage [25], with independent operators, domestic employees, and entertainment industry workers exempt from coverage requirements [26,27].” Are independent operators the same as contractors, or those who are known as contingent workers? A little more explanation is needed here. Is it possible that some of the more hazardous jobs are not covered by WSIB? What effect might that have on the estimates of national burden of leukemia, and other work-related conditions?
Response: Independent operators consist of those in sole-proprietorships or other workers employed as contractors. In this section, we noted that these workers are still eligible for WSIB coverage, as existing policies only preclude the mandatory coverage for workers that meet this criterion [Methods, Page 2]. We added a sentence to the limitations section to provide examples of independent operators, including sole-proprietorship transportation workers and contractors. Moreover, we stated how these independent operators, who are not required to have WSIB coverage, may have attenuated the effect estimates in some of our findings [Discussion, Page 13].
Comment: A question on table 2, for the construction group Construction Trades (87), there are three subgroups: Excavating, Grading, Paving and Related (871); Electrical Power Lighting and Wire Communications (873); and Construction Trades Occupations – Other (878/9). Are these the only 3 subgroups included? That is, it seems like the construction trades occupations – other group is much larger than the other 2 specific groups, is this the way the information is presented in the OCDD? If so that seems to be a significant limitation as it groups together a very diverse set of jobs, activities and potential exposures in the “other “. If this is the case, it should be mentioned as a limitation.
Response: We thank the reviewer for their comment. There are three major-level groups (intermediate level) for construction trades. However, our lost time compensation claims data also included each worker’s minor level classification (most detailed level). We decided not to report this detailed data in the form of a table due to small case counts, particularly when stratifying by sex. However, several minor-level findings are discussed in the results section (Section 3.2). As several minor-level findings are included in the results section, we believe no revisions to the limitations section is necessary.
Results for the 14 minor-level groups under ‘Construction Trades Occupations – Other’ are included below. These 14 groups included: ‘Foremen – Other Construction Trades’, ‘Carpenters and Related’, ‘Brick and Stone Masons and Tile Setters’, ‘Concrete Finishing and Related’, ‘Plasterers and Related’, ‘Painters, Paperhangers and Related’, ‘Insulating’, ‘Roofing, Waterproofing and Related’, ‘Pipefitting, Plumbing and Related’, ‘Structural Metal Erectors’, ‘Glaziers’, ‘Inspecting, Testing, Grading and Sampling’, ‘Labouring and Other Elemental Work’ and ‘Other Construction Trades, Not Classified Elsewhere’. Among female workers, case counts for all 14 minor-level groups were less than 6 and could not be reported. For male workers, case counts were reportable for 13 minor level groups, with the exception of insulating occupations. Elevated risks were seen among males in brick and stone masons and tile setting occupations.
Comment: In the discussion, paragraph 7 states “Benzene, which has sufficient evidence for an association with AML and limited evidence as a causative agent for CML and CLL [5], has been found in small concentrations among metal cleaning solvents [41]”. As mentioned earlier, given that some people in the cohort could have been working in the 1960s and 70s, benzene concentrations in solvents (and associated exposures) were much higher than are present later in time. This should be mentioned.
Response: In response to the reviewer’s comment, we added a small section to outline how the frequency and level of exposure to these chemical hazards may have decreased in recent decades [Discussion, Page 11]. In this revision, we particularly focused on benzene, and highlighted literature that identified decreases in benzene exposure due to improved engineering controls and a diminished use in occupational settings.
Comment: Finally a question on time trends, is there any way to look at the era in which people started work in an industry or occupation? This is a question, related to my earlier observation about trends over time in levels of occupational exposures.
Response: Regarding time-trends, we are unable to ascertain when an individual began working in a given occupation or industry. This is definitely an interesting area of study and one that we considered during the course of our analysis. Levels of benzene and other pollutants emitted from combustion engines are also likely to have decreased in recent decades. In response to these comments, we would also like to highlight how changes in legislation have reduced the occurrence of several hazardous substances in occupational settings. For instance, diazinon, an organophosphate pesticide with limited evidence for an association with leukemia per IARC, has been banned for use in residential areas since 2005. Improvements in engineering controls, such as personal protective equipment, are also likely to have helped reduce exposure to some hazardous substances. While we are aware of the value of exploring trends over time and could possibly analyze trends based on the date of cohort entry or birth, we are currently limited by the size of our cohort. Given that the ODSS is regularly updated with new administrative data, we may be able to perform such an analysis in the future.
